# Reliability of Capillary Blood Gas Measurements in Paralympic Sprinters at Rest: A Pilot Study

**DOI:** 10.3390/sports13070216

**Published:** 2025-07-04

**Authors:** Thiago Fernando Lourenço, Samuel Bento da Silva, Vinícius Miguel Zanetti, Ana Gabriela Almeida Maximo Pereira, João Vitor Ichikawa Quintella, Oriane Martins, Amaury Verissimo, Lazaro Alessandro Soares Nunes

**Affiliations:** 1Brazilian Paralympic Committee, São Paulo 04329-100, Brazilvinicius.zanetti@cpb.org.br (V.M.Z.); ana.pereira@cpb.org.br (A.G.A.M.P.); joao.quintella@cpb.org.br (J.V.I.Q.); oriane.santos@cpb.org.br (O.M.); amaury.verissimo@cpb.org.br (A.V.); 2Human Metabolism, Uniguaçu University, Foz do Iguaçu 85852-010, Brazil; lazaroalessandro@gmail.com

**Keywords:** acid-base status, athletes, reliability

## Abstract

Background: Sports scientists have increasingly used point-of-care methods for training load management, and blood gas analysis has shown promise in this area. However, the reproducibility of this method in high-performance athletes remains unproven. Objective: The aim of this study was to verify the reliability of acid-base variables at rest in high-performance Paralympic sprinters. Methods: Seven athletes participated, including four with visual impairments (class T12 and T13) and three with physical impairments. Approximately 500 µL of capillary blood was obtained from the fingertip and analyzed in triplicate (Samples 1, 2, and 3) using the Epoc System^®^ (Ottawa, ON, Canada) to measure pH, carbonic dioxide partial pressure (pCO_2_), bicarbonate ion (HCO_3_^−^), base excess (BE), hematocrit (Hct), hemoglobin concentration (Hb), creatinine (CRE), and urea concentration (URE). Results: No differences were found for any parameter (*p* > 0.05). The imprecision of the method ranged from 0.1% for blood pH to 6.1% for BE. Pearson’s analysis showed strong and significant relationships between all variables analyzed (*p* < 0.05). The degree of consistency among samples also indicated excellent reliability of measurements, ranging from 0.88 for Hb to 1.00 for URE. Conclusions: These results indicate that acid-base status measurements using point-of-care demonstrated excellent reliability in high-level athletes, supporting sports scientists and coaches for athlete training and management.

## 1. Introduction

In elite athletes, the balance between specific training-induced adaptations and recovery between sessions is crucial to minimize injury risks and enhance performance. Many tools are employed to monitor the effects of internal and/or external load applied to athletes. Notably, to monitor internal load, the rate of perceived exertion, heart rate, lactate concentration, and certain biomarkers are well documented in sports science [1]. Proteins, metabolites, electrolytes, and other molecules are also increasingly utilized by sports team scientists as biomarkers.

To assess skeletal muscle status (i.e., muscle damage), the most common biomarkers remain creatine kinase and urea due to their cost-effectiveness and daily applicability [2,3]. However, molecules related to endocrine regulation of muscle repair, muscle excitability, and metabolic homeostasis have also gained attention [4]. Biomarker monitoring should occur at multiple time points throughout training, the off-season, and competition cycles. For chronic changes across a season, athletes may be tested every 4–6 weeks or at key training transition points [1,5].

Recently, sports scientists have increasingly used point-of-care (POC) methods to identify and analyze reliable biomarkers that aid in making rapid, critical decisions regarding training load management. Among these biomarkers, a variety of novel molecules (e.g., CD163, heat shock proteins [HSPs], cell-free DNA [cfDNA], blood cell ratios) are suggested to add value in this context. However, the cost and effort involved in measuring these parameters regularly remain high, limiting their convenience for monitoring purposes [6,7]. In contrast, evidence of emerging biomarkers related to the immune system, such as leukocytes and pro- and anti-inflammatory cytokines, is improving [8,9]. According to Haller et al. [7], the potential use of these proteins as biomarkers in exercise settings is of particular interest, as immunological markers indicate differential disturbances in physiological homeostasis or tissue integrity. Additionally, markers of oxidative stress (i.e., malondialdehyde, protein carbonyls, and antioxidant enzymes) have also been studied due to their close connection to the immune system. High-intensity exercise is known to lead to the excessive production of reactive oxygen species (ROS) through mitochondrial electron leakage or intracellular enzymes, such as NADPH oxidases, xanthine oxidase, and phospholipase A2 [10]. ROS production induces changes in the normal physiological environment of skeletal muscle fibers and vascular endothelial function, which play a central role in the inflammatory process [11]. Nevertheless, there are few studies investigating biomarkers in Paralympic athletes. Of the few studies presented in the literature, all of them are associated with salivary assessment (cortisol, testosterone, cortisol: testosterone ratio, or secretory immunoglobulin A) [12,13,14].

In a study of four Paralympic swimmers, Sinnot-O’Connor et al. [13] found significant increases in salivary markers associated with two weeks of intensified training load (38.3%), with a subsequent decrease after a 49.5% decrease in training load. These data suggest a higher risk of upper respiratory tract infections in these athletes when submitted to intensive and prolonged training. In addition to upper respiratory tract infections, the training intensification can also induce mitochondrial impairments. Flockhart et al. [15] progressively increased high-intensity-interval-training (HIIT) loads over four weeks in healthy subjects, where, during the first three weeks, the frequency of exercise sessions increased, and during the fourth week, the training load was reduced to allow for recovery. As a result, the authors observed reduced ROS emission, closely linked to a decrease in mitochondrial respiration after the fourth training week. This could be a compensatory mechanism to counteract increases in non-mitochondrial ROS production as a protective strategy against oxidative stress. Similarly, Cardinale et al. [16] found that endurance athletes exhibited an increase in several mitochondrial autophagy and density markers, cytosolic proteins, and a decrease in mitochondrial respiration (20%) and aconitase activity, indicating reduced mitochondrial quality following four weeks of intensified training. This may also explain the increase in injury incidence associated with HIIT programs in the 2000s after its popularization. Rynecki et al. [17] found a 144% increase in all injuries, including a 137% increase in lower extremity injuries, which may be due to an unbalance between training load and recovery time.

Aconitase, commonly used to assess mitochondrial adaptation, is an enzyme that catalyzes the reversible isomerization of citrate and isocitrate and is part of the citric acid cycle (TCA) and is present in both mitochondria and cytosol. The TCA is a series of reactions in a closed loop that is closely connected to carbon dioxide production and the bicarbonate ion in skeletal muscle and red blood cells [18,19,20]. This suggests a possible connection between mitochondrial function and acid-base status. The assessment of acid-base status is commonly used in inpatient settings to diagnose gas exchange abnormalities in various pulmonary and non-pulmonary diseases, and over the past few decades, technological advancements have enabled the increasing use of point-of-care (POC) tests in clinical settings, allowing for the analysis of blood gas, electrolytes, and metabolites from capillary blood samples, in addition to arterial and venous blood. Compared to venipuncture, capillary blood can be easily obtained by pricking the skin of a fingertip or ear lobe [21]. Many studies have shown good reliability between fingertip and venous or arterial blood gas analysis in many devices, suggesting that these devices can be used as a viable alternative to arterial blood gas levels [22,23,24].

The use of blood gas analysis in exercise has gained increasing attention due to its ability to provide precise insights into an acid–base status, respiratory function, and metabolic adaptations [25,26]. Although promising, there are still scarce studies associating arterial, venous, or even capillary blood gas variables in high-level athletes. Martínez et al. [27] and Lucía et al. [28] found no differences in capillary blood gas markers in well-trained cyclists after an 8-week off-season period (↓volume and intensity of training). Recently, Lourenço et al. [29] showed that resting acid-base status can be a useful indicator for endurance performance, as they found strong relationships between blood bicarbonate ion concentrations, ventilatory threshold parameters (ventilatory threshold and respiratory compensation point), and 10 km performance.

Despite this, reproducibility studies serve as a foundational step to distinguish true physiological adaptations from potential measurement noise [30]. This is particularly relevant in high-performance and Paralympic sports, as assessing blood gas variables can not only support accurate monitoring protocols but also contribute to evidence-based decision-making in both training and medical care. To our knowledge, few studies have shown reliability data in athletes, and the current lack of studies on this topic impairs the application of capillary blood gas analysis.

We think that the use of capillary blood gas analysis in sports science—particularly within Paralympics—may offer significant advantages due to its ease of application, rapid diagnostic capability, minimal discomfort, and non-invasive sampling method. To contribute to this scenario and provide important sensitivity information for trainers and scientists, the aim of the study was to verify the intra- and interday reliability of acid-base variables at rest in high-performance Paralympic sprinters. We hypothesize that their impairments should trigger some blood acid-base alteration in comparison to non-athletes’ subjects, which may generate different interpretations for this population.

## 2. Methods and Materials

### 2.1. Participants

Seven high-level paralympic sprinters (four men and three women) participated in this study, including four athletes with visual impairments (class T12 and T13) and three with physical impairments. All participants had more than ten years of experience in training, with at least seven training sessions per week, and were members of the national athletic team in the Paralympic Games in Paris 2024 (PG), and five of them were medalists in this event. Given the limited availability of elite athletes in Paralympic Track and Field in our Training Centre, we have utilized a small but relevant sample for this exploratory investigation. The participants’ characteristics are presented in Table 1.

### 2.2. Procedures

To investigate intra-day reliability, blood samples were taken at rest 24 h after the last training session. During this period, the athletes remained without training sessions and with their diet unchanged. The analysis was performed two weeks post-PG, during which time the athletes had not engaged in any training sessions. The athletes were positioned in a seated position in an air-conditioned room at 21° C. Approximately 500 µL of fresh capillary blood was obtained from the fingertip using disposable lancets (Accu-Chek SoftClix^®^, Roche^®^, Mainhein, Germany) and collected in a heparinized syringe (Westmed^®^, Tucson, Az, USA). The samples were analyzed immediately using the validated device Epoc System^®^ (Siemens^®^, Ottawa, ON, Canada) in triplicate (Sample 1, Sample 2, and Sample 3) [31]. For each analysis, 90 µL of blood was dispensed into a single-use self-calibrated Epoc Test Card, following the manufacturer’s instructions.

Six weeks later, to investigate interday reliability, the same athletes were evaluated again at the beginning (D1) and five days after (D5). During this period, the athletes performed one training session per day, each lasting 40 min. The sessions included stretching, mobility exercises, and jogging on days 1 and 3, and strength training on days 2 and 4 (three exercises performed in two sets of four repetitions at 95% of 1RM). The fifth day was designated as a recovery day. Throughout this period, the athletes also maintained their diets unchanged.

The following parameters were analyzed in blood for this study and considered as dependent variables: pH (pH), carbonic dioxide partial pressure (pCO_2_), bicarbonate ion (HCO_3_^−^), base excess (BE), hematocrit (Hct), hemoglobin concentration (Hb), creatinine concentration (CRE) and urea concentration (URE). The study was conducted according to the guidelines of the Declaration of Helsinki and approved by the Human Research Ethics Committee of São Leopoldo Mandic University (no. 78476624.5.0000.5374). Informed consent was obtained from all subjects involved in the study.

### 2.3. Statistical Analysis

Descriptive data are presented as mean ± standard deviation and 95% confidence interval (IC95%) for each sample. The normality of the sample was tested using the Kolmogorov–Smirnov test. Differences between samples for each variable were tested using repeated measures analysis of variance (ANOVA) with Bonferroni post hoc tests. Effect sizes (eta squared-η^2^) for ANOVA were set according to Cohen et al. [32] as follows: 0.01 (small effect), 0.06 (medium effect), and 0.14 (large effect).

The reliability of measurements was analyzed by calculating the typical error (TE) using within-subject variations, following the recommendation of Hopkins [33]. To facilitate comparison across trials, the coefficient of analytical variation (CV_A_%) was calculated by dividing TE by the variable’s mean among the group in all repetitions. Additionally, the lower and upper confidence limits (CL_95%_) of the TE were also reported to evaluate the precision and sensitivity of the protocol.

Associations between samples was assessed using Pearson’s correlation coefficient (r), with the strength of correlation categorized according to Hopkins [33] as follow: small (0–0.30), moderate (0.31–0.49), large (0.50–0.69), very large (0.70–0.89), and almost perfect (0.90–1). The consistency among samples was evaluated by the intraclass correlation coefficient (ICC), where values <0.50 were considered low, 0.50–0.75 moderate, 0.75–0.90 good, and >0.90 excellent [34]. To complement the interday reliability, agreement between repeated measures was assessed using the Bland–Altman method. The mean difference (bias) and 95% limits of agreement (LoA), calculated as the mean difference ± 1.96 × standard deviation of the differences, were determined to evaluate systematic bias and random error between measurements. All analyses were conducted using GraphPad Prism 10.1 Software (Boston, MA, USA), with an alpha level set at *p* = 0.05. Due to the small sample size, gender was not compared in this study.

## 3. Results

### 3.1. Intraday Reliability

No differences were found between sample means for any parameter analyzed (*p* > 0.05). The descriptive values are presented in Table 2. The imprecision of the method, reported as CV_A_%, ranges from 0.1% for blood pH to 6.1% for BE (Table 2). The typical error ranged from 0.01 units for blood pH to 1.09 mmHg for pCO_2_. The effect size for ANOVA ranged from large for pH (*p* = 0.30; F = 1.34; η^2^ = 0.18), pCO_2_ (*p* = 0.09; F = 2.88; η^2^ = 0.32), HCO_3_^−^ (*p* = 0.06; F = 3.53; η^2^ = 0.37), BE (*p* = 0.10; F = 2.69; η^2^ = 0.31), Hb (*p* = 0.34; F = 1.18; η^2^ = 0.16) and URE (*p* = 0.39; F = 1.00; η^2^ = 0.14), to medium for CREA (*p* = 0.82; F = 0.22; η^2^ = 0.04).

The Pearson’s analysis (Table 3) showed strong and significant relationships between all samples for all variables analyzed (*p* < 0.05), and the degree of consistency among samples (ICC) also indicated excellent reliability of measurements, ranging from 0.88 for Hb to 1.00 for URE (Table 4).

### 3.2. Interday Reliability

One-way ANOVA indicated no statistically significant differences between D1 and D5 for any variable (all *p* > 0.05), supporting the temporal stability of measurements. The CV_A_% for interday comparison ranges from 0.22% for blood pH to 28.7% for BE (Table 5). The typical error ranged from 0.02 units for blood pH to 1.41 mmol·L^−1^ for BE. The effect size for ANOVA ranged from trivial for pH (*p* = 0.55; F = 0.61; η^2^ = 0.05), pCO_2_ (*p* = 0.94; F = 0.06; η^2^ = 0.005), HCO_3_^−^ (*p* = 0.96; F = 0.03; η^2^ = 0.003), CREA (*p* = 0.96; F = 0.03; η^2^ = 0.003) and URE (*p* = 0.92; F = 0.08; η^2^ = 0.007), small for HB (*p* = 0.78; F = 0.25; η^2^ = 0.02), to medium for BE (*p* = 0.34; F = 1.11; η^2^ = 0.09).

The Pearson’s correlations showed strong and statistically significant associations for most variables across all pairs of collections, ranging from 0.70 to 0.98 (*p* < 0.05), except for BE, which showed weak to moderate correlations of 0.12 (*p* > 0.05) (Table 6). 

The degree of consistency among the sample (ICC) analysis revealed moderate reliability for pH and HCO_3_^−^, good reliability for pCO_2_ and Hb, and excellent reliability for CREA and URE (Table 7). BE was the only variable with low reproducibility (ICC = 0.23; 95% CI [−0.172, 0.72]; *p* = 0.1464).

The Bland–Altman analysis revealed a mean bias of −0.020 for blood pH, with narrow limits of agreement (approximately ±0.05), indicating excellent agreement and minimal systematic error. For pCO_2_, the bias was close to zero, with moderate dispersion and limits of agreement extending to ±3.5. In HCO_3_^−^, the bias remained below 0.2, and LoA was ±1.2, supporting the consistency suggested by the ICC and correlation analyses. The BE demonstrated the poorest agreement, with inconsistent bias across comparisons and wide LoA (exceeding ±5.0). Mean differences for Hb were small (bias 0.1), though LoA were wider (±2.5) with acceptable agreement, and CREA showed excellent agreement, with minimal bias (<0.03) and narrow limits of agreement (±0.2), confirming the ICC results. For URE, despite a small mean bias (−0.5), the LoA ranged widely (±4.5), reflecting higher individual variation.

## 4. Discussion

The aim of the present study was to evaluate the intra- and interday reliability of acid-base variables at rest in high-level paralympic sprinters. Although a limited number of studies have examined the effectiveness of point-of-care devices to analyze acid-base status using capillary blood in a clinical setting, to the best of our knowledge, this is the first study to provide reliability data for high-performance athletes. 

Our imprecision results (Table 2 and Table 5) agreed with others [35,36,37], indicating the high reliability of these measurements for quantifying acid-base status using point-of-care devices. No significant differences were found in any variable analyzed when comparing the three samples during intra- or interday analysis, and we also found high consistency (ICC > 0.89) among samples for all blood variables analyzed, indicating excellent reliability of these parameters (Table 4). The only exception was the base excess (BE), which showed a moderate effect size (η^2^ = 0.09), suggesting slightly higher variability in this specific parameter. Nonetheless, even in this case, statistical significance was not achieved (*p* = 0.34), further supporting the consistency of the data. These results reflect high reproducibility across the repeated measurements, as the differences between conditions were statistically negligible and of limited practical relevance.

Although the subjects analyzed in our study differ from previous studies [21,23,38], our results, collected in the same device, corroborate their findings. In a cohort of 250 participants with various health conditions, including cancer, pneumonia, and metabolic acidosis, Kim et al. [21] found that, with the exception of partial pressure of oxygen (pO_2_), all other parameters showed equivalent values or strong correlations with reference methods. The error values for pH, pCO_2_, HCO_3_^−^, and Hb were ±0.08 (±0.04%), ±8.66 mmHg (±8%), ±2.63 mmol·L^−1^ (±15%), and ±1.67 g·dL^−1^ (±7%), respectively. These findings align with those of Cao et al. [23], who reported similar error values of pH (0.03%), pCO_2_ (3.97%), and Hb (1.02%).

When comparing our mean values, we observed slightly higher pH (7.49–7.55) and lower pCO_2_ (20.91–29.31) compared to previous studies [21,23,31], suggesting that high-level athletes may exhibit mild blood alkalinization at rest. These data are different from those found by Lourenço et al. [29] in high-level endurance athletes who show blood pH and pCO_2_ lower than the athletes evaluated in this study. A pCO_2_ value below 35 mmHg is commonly associated with hyperventilation and an increase in blood pH. Unfortunately, we did not measure ventilation data in the present study. Despite that, these data can possibly be justified by a significantly higher pulmonary vital capacity (i.e., total amount of air exhaled after maximal inhalation) in high-level athletes because of their increments in strength and resistance of respiratory muscles [39,40]. Exercise training can also lead to an increase in mitochondrial function [41], which could improve tissue CO_2_ production in the citric acid cycle. According to Faull et al. [42], athletes are more accurate at interpreting interoception signals, such as pCO_2_ and H^+^ alterations, allowing greater ventilatory control compared to sedentary people. This may be a consequence of repeated exposure to acidosis during training sessions, which could trigger a higher pCO_2_ loss through ventilation, increasing blood pH at rest. Further studies should be conducted to confirm these hypotheses.

Increased pCO_2_ levels can also stimulate the conversion of CO_2_ into bicarbonate ions, primarily in erythrocytes, through the catalytic action of carbonic anhydrase [20]. A recent study by Lourenço et al. [29] found higher bicarbonate concentration in high-level endurance runners compared to amateurs. Contrary to this hypothesis, the bicarbonate ion values in our study were similar to those found in the emergency room [21] but lower than those of high-level endurance athletes [29]. This discrepancy could be attributed to differences in training and physical characteristics between sprinters and endurance athletes, as well as the time of the season during which the analysis was conducted. Since our measurements were taken at the beginning of the season, athletes may have maintained the central (cerebral and ventilatory) but not peripheral (metabolic) adaptations following a period of rest [43]. Future studies should consider monitoring these parameters throughout the entire season to investigate potential training-induced adaptations in sprinters.

Another key factor in maintaining acid-base balance is hemoglobin. During the conversion of CO_2_ into bicarbonate ion, the hydrogen ions produced in this reaction are buffered by hemoglobin, which is reduced by oxygen dissociation, playing a crucial role in the buffer system. As all variables analyzed here, hemoglobin levels were highly consistent and reliable; however, we found values slightly higher than those reported in previous studies [23,24,31,38] but like those found in high-level endurance athletes [29], reflecting classic training-induced adaptations in high-level sprinters [44].

To assess the contribution of non-carbonic buffers (i.e., protein) on pH maintenance, base excess (BE) is commonly used in point-of-care analysis. BE is defined as the amount of strong acid that must be added to fully oxygenated blood to return the pH to 7.40 at a temperature of 37 °C and a pCO_2_ of 40 mmHg. Although its application in sports is rare, our data demonstrated BE as a reliable measure, consistent with reference values (±2 meq/L). Despite the rest, BE values agree with reference values; they are less reliable than other variables analyzed here. Furthermore, we found lower BE values than in endurance athletes [29], which can reflect a specific training-induced adaptation rather than methodological inconsistencies. From a practice standpoint, coaches and sports scientists should carefully use BE as an indicator of plasma protein buffer capacity, and future studies may provide more information about it in high-level athletes.

Creatine and urea are also commonly measured in point-of-care devices to assess kidney function and muscle catabolism. Like other parameters, creatinine and urea showed high reliability in our high-level athlete sample, demonstrating potential for use in training load monitoring throughout the season. Banfi and Del Fabbro [45] showed a correlation between body mass and creatinine concentration in athletes from different sports. These findings support the work of Haller et al. [7], who highlighted the value of creatinine and urea, along with creatine kinase, as markers of training load.

The predominance of small effect sizes reinforces that the blood gas parameters remained stable across the five-day interval—an important finding when considering the clinical and athletic applications in Paralympic athletes. The reliability of these measures may be essential for physiological monitoring and for informed decision-making regarding training and recovery strategies. However, despite these promising results, certain limitations of the study should be addressed in future investigations. Although none of the comparisons intraday reached statistical significance, the observed effect sizes ranged from medium to large, particularly for acid-base balance markers (e.g., pCO_2_, HCO_3_^−^, BE) and hematological variables (Hb, URE). These findings suggest potentially meaningful physiological changes that may not be captured solely through *p*-values, especially in studies with limited sample size, like ours. Therefore, the magnitude of these effects warrants further investigation in larger cohorts and may inform individualized monitoring strategies in high-performance or clinical athletic populations.

## 5. Conclusions

In conclusion, this study demonstrated that blood gas analysis at rest in Paralympic sprinters exhibited high reliability, mainly for pH, HCO_3_^−^, and CREA, which presented narrow limits of agreement and minimal bias, while BE showed wider limits of agreement, suggesting greater individual variability.

These findings also support the use of blood gas analysis as a reliable tool for monitoring physiological status in sprinters, both in clinical and performance contexts. However, caution is advised when interpreting variables such as base excess, and further studies are warranted to explore the impact of training load, time of day, and athletes’ impairment-related differences on these measurements.

## Figures and Tables

**Table 1 sports-13-00216-t001:** Descriptive characteristics of participants.

Athlete	Gender	Age (y)	Weight (kg)	Height (cm)	Personal Best 100-m (s)	Deficiency
1	Male	32	75.4	177	11.13	Transtibial amputation
2	Male	27	73.7	168	10.87	Visual impairment
3	Male	26	72.1	182	11.76	Visual impairment
4	Female	25	41.3	145	17.04	Spinal cord injury
5	Male	28	73.6	184	11.98	Visual impairment
6	Female	38	60.0	162	12.54	Arm amputation
7	Female	27	59.5	164	11.78	Visual impairment

**Table 2 sports-13-00216-t002:** Mean (± standard deviation), coefficient of variation, typical error, and confidence limits of all acid-base variables analyzed in three different blood samples.

Parameter	Sample 1		Sample 2		Sample 3		CV_A_%	Typical Error	CL_95%_
	Mean ± SD	IC95%	Mean ± SD	IC95%	Mean ± SD	IC95%			
pH (unit)	7.52 ± 0.03	7.49–7.54	7.52 ± 0.03	7.49–7.55	7.53 ± 0.03	7.49–7.56	0.1%	0.01	0.01–0.02
pCO_2_ (mmHg)	25.8 ± 4.37	21.7–29.8	25.1 ± 4.9	20.6–29.7	24.3 ± 3.7	20.8–27.8	3.8%	1.09	0.81–1.83
HCO_3_^−^ (mmol·L^−1^)	20.8 ± 2.6	18.4–23.2	20.3 ± 2.79	17.7–22.9	20.0 ± 2.3	17.9–22.1	2.5%	0.49	0.36–0.82
BE (mmol·L^−1^)	−0.41 ± 1.73	−1.02–1.19	−0.77 ± 1.81	−1.44–0.90	−0.83 ± 1.74	−1.43–0.78	6.1%	0.37	0.27–0.74
Hb (g·dL^−1^)	14.8 ± 1.2	13.7–15.9	15.0 ± 0.8	14.1–15.8	15.1 ± 1.31	13.9–16.3	1.8%	0.37	0.27–0.61
CREA (mg·dL^−1^)	1.06 ± 0.24	0.83–1.28	1.06 ± 0.29	0.79–1.32	1.04 ± 0.25	0.81–1.27	2.5%	0.04	0.03–0.06
URE (g·dL^−1^)	17.4 ± 6.5	11.4–23.4	17.2 ± 6.6	11.1–23.4	17.4 ± 6.5	11.4–23.4	0.8%	0.27	0.20–0.45

Legend: SD—standard deviation; CI 95%—mean confidence interval of 95%; CV_A_%—coefficient of variation; CL_95%_—confidence limit of 95%.

**Table 3 sports-13-00216-t003:** The correlation coefficient between three different blood samples of all blood variables analyzed.

Parameter		Sample 2	Sample 3
pH	Sample 1	0.89 *	0.91 *
	Sample 2	-	0.90 *
pCO_2_	Sample 1	0.95 *	0.92 *
	Sample 2	-	0.97 *
HCO_3_^−^	Sample 1	0.97 *	0.94 *
	Sample 2	-	0.98 *
BE	Sample 1	0.97 *	0.97 *
	Sample 2	-	0.95 *
Hb	Sample 1	0.96 *	0.87 *
	Sample 2	-	0.94 *
CREA	Sample 1	0.97 *	0.99 *
	Sample 2	-	0.98 *
URE	Sample 1	1.00 *	1.00 *
	Sample 2	-	1.00 *

Legend: * *p* < 0.05.

**Table 4 sports-13-00216-t004:** Consistency among samples for all blood variables analyzed based on absolute agreement of intraclass correlation coefficient (ICC) and 95% confidence interval (IC95%).

Parameter	ICC	Lower IC95%	Upper IC95%	F	*p*
pH	0.89	0.68	0.98	27.25	<0.001
pCO_2_	0.91	0.72	0.98	41.60	<0.001
HCO_3_^−^	0.94	0.79	0.99	66.91	<0.001
BE	0.95	0.82	0.99	68.73	<0.001
Hb	0.88	0.64	0.98	23.05	<0.001
CREA	0.97	0.90	0.99	93.78	<0.001
URE	1.00	1.00	1.00	2710	<0.001

**Table 5 sports-13-00216-t005:** Mean (± standard deviation), coefficient of variation, typical error, and confidence limits of all acid-base variables analyzed between five-day interval (D1 and D5).

Parameter	D1		D5		CV_A_%	Typical	CL_95%_
	Mean ± SD	IC95%	Mean ± SD	IC95%		Error	
pH (unit)	7.49 ± 0.03	7.49–7.54	7.50 ± 0.04	7.49–7.53	0.22%	0.02	0.01–0.03
pCO_2_ (mmHg)	29.1 ± 2.54	24.91–29.12	28.71 ± 2.43	24.32–28.68	3.49%	1.25	0.91–2.05
HCO_3_^−^ (mmol·L^−1^)	22.48 ± 1.23	18.37–22.03	22.46 ± 0.96	18.70–22.32	1.68%	0.51	0.37–0.84
BE (mmol·L^−1^)	−0.11 ± 1.52	−1.03–1.27	0.89 ± 1.78	−0.61–1.48	28.7%	1.41	1.03–2.31
Hb (g·dL^−1^)	16.01 ± 1.64	13.59–14.37	15.43 ± 1.99	14.19–16.07	4.55%	0.81	0.59–1.34
CREA (mg·dL^−1^)	1.05 ± 0.29	0.09–0.24	1.07 ± 0.22	0.11–0.44	4.76%	0.07	0.05–0.12
URE (g·dL^−1^)	19.25 ± 5.23	11.8–4.37	19.75 ± 5.95	11.6–4.15	5.10%	1.16	0.85–1.92

Legend: SD—standard deviation; IC95% —mean confidence interval of 95%; CV_A_%—coefficient of variation; CL_95_%—confidence limit of 95%.

**Table 6 sports-13-00216-t006:** The correlation coefficient between five-day interval of all blood variables analyzed.

Parameter	r
pH	0.76 *
pCO_2_	0.84 *
HCO_3_^−^	0.70 *
BE	0.12
Hb	0.90 *
CREA	0.97 *
URE	0.98 *

Legend: * *p* < 0.05.

**Table 7 sports-13-00216-t007:** Consistency among samples for all blood variables analyzed based on absolute agreement of intraclass correlation coefficient (ICC) and 95% confidence interval (IC95%) between five-day interval.

Parameter	ICC	Lower IC95%	Upper IC95%	F	*p*
pH	0.73	0.361	0.932	9.12	<0.001
pCO_2_	0.89	0.68	0.974	24.89	<0.001
HCO_3_^−^	0.75	0.388	0.936	9.82	<0.001
BE	0.23	−0.172	0.72	1.89	>0.05
Hb	0.85	0.579	0.963	17.3	<0.001
CREA	0.91	0.73	0.979	30.83	<0.001
URE	0.965	0.886	0.992	82.54	<0.001

## Data Availability

The raw data supporting the conclusions of this article will be made available by the authors on request.

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
