# Peer review of "Reliability of Capillary Blood Gas Measurements in Paralympic Sprinters at Rest: A Pilot Study"

_sports, 2025, doi:10.3390/sports13070216_

Round 1
Reviewer 1 Report
Comments and Suggestions for Authors
This is a very interesting and applicable study in the field of sports.
Introduction
The physiological basis is detailed, although more practical examples should be provided for sports, in addition to those mentioned for HIIT.
A study in which a correlation between this variable and another variable was detected could be mentioned to confirm whether it is valid.
Methods
Given the small number of participants (understandable due to the profile of the athletes), it is suggested that the title of the study be changed to a pilot study.
In the descriptive data, experience could be indicated as something important that can alter some physiological variables due to accumulated adaptation.
The type of session they had performed before collecting the sample was detailed. It is indicated only that it is light, but more details should be provided because the internal load may differ, as well as the recovery.
Accompanying this section with an illustrative photograph would be adequate to obtain an overall view of the experiment.
They should indicate the reason for not differentiating the sexes; this result could be interesting.
Discussion
In the first paragraph, you indicate that this is the first study of these characteristics. Recent studies have shown that although they do not measure acid-base, they have measured lactate and other related cardiocirculatory variables in Paralympic sports (https://doi.org/10.3390/sports12070188).
It is necessary to indicate at least two applications in the field of training because the direction of their study is related to the recovery between stimuli.
The lack of distinction between sexes or small samples should be included as limitations of the study.
Author Response
Comments 1: The physiological basis is detailed, although more practical examples should be provided for sports, in addition to those mentioned for HIIT.
Response 1: Thank you for your suggestions and we add in line 65:
In four Paralympic swimmers, Sinnot-O’Connor et al. [13] found significant increases in salivary markers associated to a two weeks of intensified training load (38.3%) with a subsequential decrease after 49.5% decrements in training load. These data suggest a higher risk of upper respiratory tract infections in these athletes when submitted to intensive and prolonged training. In addition to upper respiratory tract infections, the training intensification can also induce mitochondrial impairments.
Comments 2: A study in which a correlation between this variable and another variable was detected could be mentioned to confirm whether it is valid.
Response 2: Thank you for your suggestions and we add in line 100:
The use of blood gas analysis in exercise has gained increasing attention due to its ability to provide precise insights into an acid–base status, respiratory function, and metabolic adaptations [25,26]. Although promising, are still scarce studies associating arterial, venous or even capillary blood gas variables in high-level athletes. Martínez et al. [25] and Lucía et al. [26] found no differences on capillary blood gas markers in well-trained cyclists after 8 weeks off-season period (↓volume and intensity of training). Recently, Lourenço et al. [27]showed that resting acid-base status can be a useful indicator for endurance performance, once strong relationships between blood bicarbonate ion concentrations, ventilatory threshold parameters (ventilatory threshold and respiratory compensation point) and 10-km performance were found.
Comments 2: Given the small number of participants (understandable due to the profile of the athletes), it is suggested that the title of the study be changed to a pilot study.
Response 3: We agree with this suggestions and we add it. However, in response to reviews 2, we also add a comparison interday in this study. We really appreciate your opinion in keep the "pilot study" on title or no.
Comments 3: In the descriptive data, experience could be indicated as something important that can alter some physiological variables due to accumulated adaptation.
Response 4: Thank you for the suggestion. It is added in line 130.
All participants had more than ten years of experience in training and were members of the national athletic team in Paralympic Games in Paris 2024 (PG) and five of them were medalists in this event.
Comments 4: The type of session they had performed before collecting the sample was detailed. It is indicated only that it is light, but more details should be provided because the internal load may differ, as well as the recovery.
Response 4: We appreciate this suggestions and is added in line 148:
Six weeks later, to investigate interday reliability, the same athletes were evaluated again in the beginning (D1) and five days after (D5). During this period, the athletes performed one training session per day, each lasting 40 minutes. The sessions included stretching, mobility exercises, and jogging on days 1 and 3, and strength training on days 2 and 4 (three exercises performed in two sets of four repetitions at 95% of 1RM). The fifth day was designated as a recovery day.
Comments 5: Accompanying this section with an illustrative photograph would be adequate to obtain an overall view of the experiment.
Response 5: We thank your suggestion, however we decide to not include in the text because this device is already well known and there are many pictures in traditional search websites.
Comments 6: They should indicate the reason for not differentiating the sexes; this result could be interesting.
Response 6: We thank your suggestions and it is added in line 184:
Due the sample’s size we did not compare gender in the study.
Comments 7: In the first paragraph, you indicate that this is the first study of these characteristics. Recent studies have shown that although they do not measure acid-base, they have measured lactate and other related cardiocirculatory variables in Paralympic sports (https://doi.org/10.3390/sports12070188).
Response 7: We appreciate your comment, but the study cited in comment measures only mechanic variables. Maybe there was a mistake on the suggestion. If we are correct, please let us know about the correct study.
Comments 8: It is necessary to indicate at least two applications in the field of training because the direction of their study is related to the recovery between stimuli.
Response 8: We thank the suggestions and this issue is added in line 337:
The predominance of small effect sizes reinforces that the blood-gas parameters remained stable across the five-day interval — an important finding when considering the clinical and athletic applications in Paralympic athletes. The reliability of these measures may be essential for physiological monitoring and for informed decision-making regarding training and recovery strategies. However, despite these promising results, certain limitations of the study should be addressed in future investigations.
Comments 9: The lack of distinction between sexes or small samples should be included as limitations of the study.
Response 9: We thank the suggestions and this issue is added in line 345:
These findings suggest potentially meaningful physiological changes that may not be captured solely through p-values, especially in studies with limited sample size, like our.
Reviewer 2 Report
Comments and Suggestions for Authors
- Introduction
The research addresses a novel and relevant topic: the reliability of capillary blood gas measurement in high-performance Paralympic sprinters. This topic is important because these biomarkers can be used for monitoring training load and physiological adaptation. The rationale of the study is well stated and highlights the lack of evidence on the reproducibility of these methods in elite athletes. It also highlights the importance of blood gas analysis in the clinical setting and its potential application in decision-making in sport.
However, further contextualisation is lacking, as studies on biomarkers in athletes are mentioned, but it is not explained why capillary blood gas analysis is particularly relevant in Paralympic sprinters. Literature on physiological differences in this population could be added. The sample size is also not justified, as only seven athletes were analysed, but it is not explained why this sample is sufficient to assess the reliability of the method. It is recommended to add a reference to support the number of participants or discuss the limitation. Finally, the authors do not provide a clearer and more structured hypothesis, as only the general aim of the study is mentioned, but no explicit hypothesis on what is expected to be found.
- Methodology
The authors used an appropriate experimental design as their methodology, with triplicate measurements to assess reliability. They also used the Epoc System®, a validated device for blood gas measurement in clinical and sports settings. They also reported ethical approval and informed consent, complying with human research regulations.
However, this section lacks details about the sample: The athletes' level of training beyond their Paralympic classification is not described. It is recommended to include information on their training volume and frequency. There is also an absence of a control group: Only Paralympic sprinters were evaluated without a comparison with conventional athletes or a control group, which could have provided a better perspective on the reliability of the method. Finally, the authors also fail to mention the level of pre-measurement standardisation: Were variables such as diet, hydration or exposure to pre-test environmental conditions controlled for? Including this information would improve the validity of the study.
- Results
In reference to the results section, the authors present the results clearly in well-structured tables, facilitating the interpretation of the data. They also calculated coefficients of analytical variation (CVA%), which allows the accuracy of the measurements to be assessed, as well as Pearson's correlation coefficient and the intraclass correlation coefficient (ICC) to assess the reliability of the measurements.
However, there is a lack of more detailed interpretation of the reliability values. Although high ICC values (0.88-1.00) are presented, there is no discussion of what values would be considered acceptable in this context. It is recommended to add references explaining the meaning of these values in reliability studies.
I also perceive, possible errors in Table 2, as typical error values and confidence limits are observed that might require revision, especially in BE and pCO₂, where the ranges seem wide. Verification of these calculations is recommended. Finally, I do not appreciate comparison with other studies: Reliability coefficients are reported, but not compared with previous studies of capillary gas analysis in athletes. Including a comparison would help to contextualise the relevance of the findings.
- Discussion
In this section, the authors relate the findings to previous studies, providing context on the reliability of the method. They also propose a possible physiological explanation for the observed differences in pCO₂ concentration and pH in elite athletes compared to clinical reference values. They also recognise the importance of assessing acid-base status in sports performance and its potential for monitoring training load.
However, the authors present an over-generalisation of the results. It is suggested that the method can be applied to other athletes without testing different sporting populations. It is recommended to moderate this statement and suggest further studies in other groups. Similarly, possible methodological biases are not discussed, as the authors do not mention whether familiarity with the blood collection technique may have affected the accuracy of the measurements. It is recommended that this reflection be included. Finally, the practical applicability of the findings is also not explored: It is mentioned that the values may be useful for coaches and sport scientists, but it is not explained how these results could be integrated into training planning.
- Conclusions
With reference to the conclusions, the authors summarise the findings in a clear way, highlighting the high reliability of capillary blood gas measurements in Paralympic athletes. Furthermore, they mention the need for future studies to validate these results in different sporting contexts.
However, in line with my comment in the discussion section, the conclusions are too optimistic. It is suggested that this method can be widely used in training monitoring without acknowledging the limitations of the study (small sample size, lack of comparison with other athletes).
Finally, there is an absence of concrete recommendations for future research: The importance of further study of this method is mentioned, but no specific directions for future research are proposed (e.g., assessing reliability during exercise rather than only at rest).
All in all, the research presents a valuable contribution on the reliability of capillary blood gas analysis in Paralympic sprinters. However, to strengthen the manuscript and to anticipate possible questions from the Editor, it is recommended:
- Better justify the choice of sample size.
- Include more discussion on the practical applicability of the results.
- Better contextualise the findings within previous studies in elite athletes.
- Further discuss the limitations of the study, including possible biases.
- Review and verify the values reported in the tables to avoid typographical errors or inconsistencies.
Author Response
Comments 1: ... further contextualisation is lacking, as studies on biomarkers in athletes are mentioned, but it is not explained why capillary blood gas analysis is particularly relevant in Paralympic sprinters. Literature on physiological differences in this population could be added.
Response 1: We really thank the comments and we try to improve this issue on introduction. We work harder in the introduction to contemplate these suggestions.
Comments 2: The sample size is also not justified, as only seven athletes were analysed, but it is not explained why this sample is sufficient to assess the reliability of the method. It is recommended to add a reference to support the number of participants or discuss the limitation.
Response 2: Thank you for this observation. Despite the sample being made up of high-level Paralympic sprinters, who are part of a small population worldwide, we also received this questions from other reviewer that suggested to modify the study to an "pilot study". We include it on title: RELIABILITY OF CAPILLARY BLOOD GAS MEASUREMENTS IN PARALYMPIC SPRINTERS AT REST: A PILOT STUDY
Comments 3: Finally, the authors do not provide a clearer and more structured hypothesis, as only the general aim of the study is mentioned, but no explicit hypothesis on what is expected to be found.
Response 3: We appreciate the suggestions and it are added in line 122: We hypothesize that their impairments should trigger some blood acid-base alteration in comparison to non-athletes’ subjects which may generate different interpretations for this population.
Comments 4: However, this section lacks details about the sample: The athletes' level of training beyond their Paralympic classification is not described. It is recommended to include information on their training volume and frequency. There is also an absence of a control group: Only Paralympic sprinters were evaluated without a comparison with conventional athletes or a control group, which could have provided a better perspective on the reliability of the method. Finally, the authors also fail to mention the level of pre-measurement standardisation: Were variables such as diet, hydration or exposure to pre-test environmental conditions controlled for? Including this information would improve the validity of the study.
Response 4: We appreciate all the suggestions and the first one are added in line 127: Seven high-level paralympic sprinters (four men and three women) participated in this study, including four athletes with visually impairments (class T12 and T13) and three with physical impairment.
Regarding to the control group, since we are linked to the Brazilian Paralympic Committee, we do not have access to athletes without disabilities to make this comparison. We have added suggestions for future studies and the limitations of the present study in the text as a way of answering these questions.
Regarding to the pre-measurement standardization, are added in lines 138 and 153: To investigate intra-day reliability blood samples were taken at rest 24 hours after the last training session. During this period the athletes remained without training sessions and with their diet unchanged. / Throughout the study analysis period, the athletes maintained their diets unchanged.
Comments 5: Although high ICC values (0.88-1.00) are presented, there is no discussion of what values would be considered acceptable in this context. It is recommended to add references explaining the meaning of these values in reliability studies.
Response 5: Thank you for this observation. We agree with it and tried to improve this explanation in line 268.
Comment 6: I also perceive, possible errors in Table 2, as typical error values and confidence limits are observed that might require revision, especially in BE and pCO2, where the ranges seem wide. Verification of these calculations is recommended.
Response 6: We appreciate this observation and we have verified all calculations and we have used the Hopkins spereadsheet (https://sportsci.org/2015/ValidRely.htm). The same data were found after this procedure.
Comments 7: Finally, I do not appreciate comparison with other studies: Reliability coefficients are reported, but not compared with previous studies of capillary gas analysis in athletes. Including a comparison would help to contextualise the relevance of the findings.
Responde 7: We thank the observation. In fact, it is not to our knowledge studies that have shown reliability data in athletes. We tried to improve this issue in line 100.
Comments 8: However, the authors present an over-generalisation of the results. It is suggested that the method can be applied to other athletes without testing different sporting populations. It is recommended to moderate this statement and suggest further studies in other groups. Similarly, possible methodological biases are not discussed, as the authors do not mention whether familiarity with the blood collection technique may have affected the accuracy of the measurements. It is recommended that this reflection be included. Finally, the practical applicability of the findings is also not explored: It is mentioned that the values may be useful for coaches and sport scientists, but it is not explained how these results could be integrated into training planning.
Response 8: Maybe it was the main suggestions made to our study which generates an huge difference in the discussion section. We tried to incorporate this suggestion during the entire text and we hope to succeed.
Comments 9: However, in line with my comment in the discussion section, the conclusions are too optimistic. It is suggested that this method can be widely used in training monitoring without acknowledging the limitations of the study (small sample size, lack of comparison with other athletes).
Response 9: Thank you for this suggestion. We tried to incorporate it in line 343: However, despite these promising results, certain limitations of the study should be addressed in future investigations. Although none of the comparisons intraday reached statistical significance, the observed effect sizes ranged from medium to large, particularly for acid-base balance markers (e.g., pCO₂, HCO₃⁻, BE) and hematological variables (Hb, URE). These findings suggest potentially meaningful physiological changes that may not be captured solely through p-values, especially in studies with limited sample size, like ours. Therefore, the magnitude of these effects warrants further investigation in larger cohorts and may inform individualized monitoring strategies in high-performance or clinical athletic populations.
Comments 10: Finally, there is an absence of concrete recommendations for future research: The importance of further study of this method is mentioned, but no specific directions for future research are proposed (e.g., assessing reliability during exercise rather than only at rest).
Response 9: Thank you for this suggestion. We also tried to incorporate it in line 358: However, caution is advised when interpreting variables such as base excess and further studies are warranted to explore the impact of training load, time of day, and athletes’ impairment-related differences on these measurements.
Reviewer 3 Report
Comments and Suggestions for Authors
Dear authors,
Overall a straight forward paper and methods and results align with the aims of the paper. The biggest concern I have and the authors need to do a better job at is explaining the rational on why Paralympic sprinters at rest are any different and need a test of reliably. To take blood at a single time point and then test to see if that blood is the same has to do with more of the reliably of the testing equipment not the person itself. Can a machine test the same person and get the same results, it doesn’t matter if that person is young, old, elite athletes or sedentary, the test is of the machine which is already proven reliable and valid. A better case needs to be made for why in this population needs a reliability study. Usually this type of reliability study is part of a methods of a larger study.
Author Response
Comments 1: The biggest concern I have and the authors need to do a better job at is explaining the rational on why Paralympic sprinters at rest are any different and need a test of reliably. To take blood at a single time point and then test to see if that blood is the same has to do with more of the reliably of the testing equipment not the person itself.
Response 1: We really appreciate and thank these comments which gave us an opportunity to change the study. We hope that this "new study" meets all expectations.
Round 2
Reviewer 3 Report
Comments and Suggestions for Authors
Authors have done a excellent job addressing my concerns as well as concerns of the other reviews. The manuscript is substantially better and has a clearer focus and aim. The major changes that have been corrected make this paper more of a novel purpose as it is more of a proof that these measurements work in this specific population. Overall a better more revised manuscript.